# Design of a Multi-Epitope Vaccine against *Histoplasma capsulatum* through Immunoinformatics Approaches

**DOI:** 10.3390/jof10010043

**Published:** 2024-01-05

**Authors:** Pedro Henrique Marques, Sandeep Tiwari, Andrei Giacchetto Felice, Arun Kumar Jaiswal, Flávia Figueira Aburjaile, Vasco Azevedo, Mario León Silva-Vergara, Kennio Ferreira-Paim, Siomar de Castro Soares, Fernanda Machado Fonseca

**Affiliations:** 1Postgraduate Interunits Program in Bioinformatics, Federal University of Minas Gerais, Belo Horizonte 31270-901, Brazil; pedromarqueshbio@gmail.com (P.H.M.); arunjaiswal1411@gmail.com (A.K.J.); 2Department of Preventive Veterinary, Medicine, School of Veterinary Medicine, Federal University of Minas Gerais, Belo Horizonte 31270-901, Brazil; faburjaile@gmail.com; 3Institute of Biology, Federal University of Bahia, Salvador 40170-115, Brazil; sandip_sbtbi@yahoo.com; 4Institute of Health Sciences, Federal University of Bahia, Salvador 40170-115, Brazil; 5Department of Microbiology, Immunology and Parasitology, Federal University of Triangulo Mineiro, Uberaba 38015-050, Brazil; andreigf@hotmail.com (A.G.F.); siomars@gmail.com (S.d.C.S.); 6Department of Genetics, Ecology and Evolution, Federal University of Minas Gerais, Belo Horizonte 31270-901, Brazil; vascoariston@gmail.com; 7Department of Infectious Diseases, Federal University of Triangulo Mineiro, Uberaba 38025-440, Brazil; marioleon1956@gmail.com; 8Department of Biomedicine, Federal University of Triangulo Mineiro, Uberaba 38025-350, Brazil

**Keywords:** histoplasmosis, immunoinformatic, fungi vaccine, multi-epitope, epizootic lymphangitis

## Abstract

Histoplasmosis is a widespread systemic disease caused by *Histoplasma capsulatum*, prevalent in the Americas. Despite its significant morbidity and mortality rates, no vaccines are currently available. Previously, five vaccine targets and specific epitopes for *H. capsulatum* were identified. Immunoinformatics has emerged as a novel approach for determining the main immunogenic components of antigens through in silico methods. Therefore, we predicted the main helper and cytotoxic T lymphocytes and B-cell epitopes for these targets to create a potential multi-epitope vaccine known as HistoVAC-TSFM. A total of 38 epitopes were found: 23 common to CTL and B-cell responses, 11 linked to HTL and B cells, and 4 previously validated epitopes associated with the B subunit of cholera toxin, a potent adjuvant. In silico evaluations confirmed the stability, non-toxicity, non-allergenicity, and non-homology of these vaccines with the host. Notably, the vaccine exhibited the potential to trigger both innate and adaptive immune responses, likely involving the TLR4 pathway, as supported by 3D modeling and molecular docking. The designed HistoVAC-TSFM appears promising against *Histoplasma*, with the ability to induce important cytokines, such as IFN-γ, TNF-α, IL17, and IL6. Future studies could be carried out to test the vaccine’s efficacy in in vivo models.

## 1. Introduction

*Histoplasma capsulatum*, a globally distributed intracellular pathogenic fungus, is notably found in regions such as the Mississippi and Ohio river valleys in the USA, Central and South America, Africa, and Australia [1,2,3,4]. This systemic mycosis, known as histoplasmosis, affects the respiratory system of healthy individuals and those with compromised immune systems. It occasionally affects other animals, such as horses, dogs, and cats. This pathogen can be found in various environmental niches, including caves, nitrogen-rich habitats, and in bat and bird droppings, where it exists in its mycelial form, producing both micro- and macroconidia. Once these structures are released into the air, they become inhalable and can reach the respiratory system of potential hosts, which serves as the primary route of infection. In response to the local temperature shift, the microorganism transforms into a yeast morphology, the most virulent form within the host [5,6].

The interaction between the pathogen and the host is crucial in determining the course of the infection. Factors such as the immune response, the quantity of inhaled micro- and macroconidia, and the virulence of the strain can significantly influence the outcome of disease. In acute infection, a flu-like condition typically emerges within a few weeks [7]. This condition can also spread, particularly among immunocompromised patients with low counts of CD4+ cells, resulting in a reduced lung capacity and damage to other organs [8]. Furthermore, differential diagnosis becomes imperative, given its potential to be mistaken for viral or bacterial infections like tuberculosis [9]. The pathogen has a large number of virulence factors, including the Hsp60 protein, which favor the cellular invasion in the innate immune system, allowing its resistance and dissemination from the site of infection [10]. In animals, clinical manifestations differ, encompassing pulmonary and systemic infections, lymphadenopathy, and dissemination to the skin and bones, mainly observed in primates. The disease is recognized as epizootic lymphangitis among horses and remains one of the leading causes of mortality in this species [11].

Histoplasmosis is highly prevalent in some regions of the United States in North America, although cases are still underdiagnosed. In Brazil, the prevalence of disease has been increasing, especially in immunocompromised patients, over the last few years [12,13,14].

The yeast of *H. capsulatum* is highly capable of surviving inside macrophages, which does not occur with other cell types. Although several cells can stamp out the pathogen, some lung-resident macrophages are unable to eliminate the internalized yeasts due to the impediment of lysosomal acidification and phagolysosomes fusion by the fungus. This allows for its development inside the phagocytic cell until apoptosis occurs, thus leading to dissemination to other cells [5]. On the other hand, dendritic cells appear as the primary communication between the innate and adaptive immune response in *H. capsulatum* infection, presenting antigens and generating mainly an immune response of CD8+ and CD4+ T lymphocytes. Notably, the initial infection polarizes a Th1-type response through the production of cytokines such as IL-6, IL-12, and IL-23, which results in Th1 cells with a high production of IFN-y and TNF-a cytokines (especially the latter in recurrent infections) [15]. Thus, T lymphocytes are primarily responsible for the containment and elimination of the pathogen, both by the development of Th1 helper cells and by the cytotoxicity of CD8+ T cells. In addition, the Th17 pattern of response can be concomitantly activated through IL-17 and IL-23 and is essential in regulating the immune response as well as to assist in elimination of the agent. On the other hand, the role of antibodies and the humoral immune response have not been fully elucidated, and high levels of antibodies do not necessarily indicate protection against the disease [16].

Attempts to identify antigens and epitopes able to promote an effective and long-lasting immune response against histoplasmosis have been made in the past. However, no vaccine has been developed. Previously, immunoproteomic analyses have highlighted the M antigen of *H. capsulatum* as a specific target of interest for vaccine production [17] once its expression transfers to the cell surface, and this has already been extensively studied. In addition, another study identified and synthesized four epitopes from HSP60, an Enolase, and a protein called HSC82, and all these induced the proliferation of CD4+ and CD8+ T cells with Th1 and Th17 patterns, with high levels of IFN-γ, IL-17, and IL-2 being produced [18]. Thus, they were established as potential epitopes for future development of a vaccine. They could also lead to cross-protection against other fungal pathogens since they have high similarity with different epitopes from other species. Finally, we have recently conducted one of the only studies on subtractive genomics and reverse vaccinology for fungi, where four vaccine targets were identified as capable as acting as antigens that can be used in the production of new vaccines against *Histoplasma capsulatum* [19]. In this context, it is possible to predict epitopes that elicit immune responses associated with T and B lymphocytes and to develop new and effective vaccines [20,21].

Furthermore, recent global climate change has expanded the geographic niche of *H. capsulatum*, modifying the behavioral patterns of birds and bats, and increased temperatures and ultraviolet exposure, which are also selective factors for the more virulent strains. High exposures to *H. capsulatum* will generate a future trend of new cases and deaths of humans and animals by histoplasmosis, and thus, to avoid such problems, new treatments and prevention methods must be found. However, it is worth highlighting that the immunoinformatic approach has some difficulties when applied to fungi. For example, there are no multi-epitope vaccines against fungi tested in animal models yet. Thus, we designed a new vaccine entitled HistoVAC-TSFM through immunoinformatics, capable of promoting an effective and long-lasting T lymphocyte immune response with probable induction of Th1 and Th17 patterns, essential for the prevention of histoplasmosis. In the future, new studies could test its efficacy in animal models and then in humans.

## 2. Materials and Methods

### 2.1. Retrieval of Antigenic Sequences

A total of five potential antigens were used for epitope prediction. From these, four have been previously identified through reverse vaccinology (RV) analysis [19] and were considered potential vaccine targets. They have the respective protein IDs EEH10718.1, EEH11056.1, EEH04925.1, and EEH09124.1. Their sequences and one from antigen M (AAB84182.2) were retrieved from NCBI public databases (https://www.ncbi.nlm.nih.gov/ (accessed on 15 May 2023)). In addition, epitopes previously analyzed by Kischkel et al. in 2021 [18] that demonstrated to be highly immunogenic were added to the end of the epitope prediction: “FENLGARL” and “ILGDIGILTNATVFT”, belonging to HSP60, “KPYVLPVPF” from Enolase, and “FAERIHKLVSLGLNI” from the HSC82 protein.

### 2.2. MHCI, MHCII, and B-Cell Epitope Prediction, Filtering, and Overlapping

MHCI and MHCII epitopes were predicted using two independent tools from the IEDB (Immune Epitope Database) for higher accuracy [22]. For MHCI, the “Recommended” and “netmhccons” methods were used to predict nine amino acid epitopes, while for MHCII, “Recommended” and “nn_align” were used to predict 15 amino acid epitopes (standard size), both from a collection of 27 MHCI and MHCII reference alleles. In this way, using two different tools, it was possible to achieve greater accuracy and precision in prediction [21]. In addition, B-cell epitopes of size 16 were predicted by the ABCpred tool [23] and only those with a score better than 0.51 were selected for the next steps. Initially, only MHCI and MHCII epitopes with an IC50 less than 50 and a percentile rank less than 1 were chosen. However, the analysis of MHCI epitopes revealed numerous MHCI epitopes with an IC50 less than 50. Therefore, to select only those with the highest probability of generating the best possible immune response, the IC50 filtering value for MHCI was reduced to ten.

After that, the epitope overlap analysis was first performed using house python scripts to select only the MHCI epitopes predicted simultaneously by the “Recommended” method and in “Ann”, and for MHCII the common ones between “Recommended” and “nn_align” were selected. Secondly, the same overlap analysis was performed between MHCI epitopes (9 residues) and B-cell epitopes (16 residues) and between MHCII epitopes (15 residues) and B-cell epitopes, keeping those of smaller size. Thus, the final epitopes that compose the vaccine sequence are those capable of simultaneously inducing MHCI and B cells and MHCII and B cells. Finally, the MHCI immunogenicity tool was used to predict only the epitopes with an immunogenicity for MHCI greater than 1, filtering out the rest [24].

### 2.3. Cluster Analysis, Epitope Conservation between Strains, and Population Coverage

To select only one remarkably similar epitope (those with none or up to two amino acid difference), a clustering analysis was performed using the IEDB tool “Epitope Cluster Analysis”, thus eliminating duplicated or similar epitopes. In the development of a vaccine, the immunogen is expected to be able to protect against all strains of the species. However, mutational and recombination events may provide new variants. Thus, to verify the epitope coverage among *H. capsulatum* strains, the IEDB tool “Epitope Conservancy Analysis” [25] was used together with the complete reference genomes of the *H. capsulatum* (H88, WU24, and G186) species available in the NCBI database in fasta format (https://www.ncbi.nlm.nih.gov/ (accessed on 15 May 2023)). The tool aligns the epitope and the specific lineage protein for each analyzed epitope, calculating the similarity percentage.

Finally, genetic variability at MHC (Major Histocompatibility Complex) loci results in different frequencies of MHC molecules expressed in other ethnic groups. This poses a significant challenge for developing a vaccine with global coverage. In this context, to assess the feasibility and efficacy of the proposed vaccine, the IEDB population coverage analysis tool was used. This tool calculates the percentage of individuals in a specific region likely to show an adequate immune response to the selected epitopes [26]. In this way, it was possible to assess the overall quality of the vaccine. For this analysis, all relevant alleles used in epitope prediction were thoroughly evaluated for each final epitope chosen, thus ensuring a comprehensive and accurate assessment of the potential population coverage of the vaccine.

### 2.4. Evaluation of the Immunogenicity of MHCI Epitopes for Dogs and Horses

In addition to humans, dogs and horses are frequently affected by histoplasmosis, and new forms of prevention are needed for these animals. In addition, an effective vaccine can help reduce the spread of histoplasmosis in endemic areas, contributing to veterinary public health. The availability of a safe and efficient vaccine can provide dog and horse owners with a reliable preventive measure, promoting the health and well-being of these animals. For this reason, the final MHCI epitopes for humans were evaluated and predicted by the IEDB “Recommended” method for their ability to induce responses associated with canine and equine MHCI alleles to be used in future animal vaccines or even to test whether the present human vaccine could be applicable to these animals, thus becoming a universal vaccine [27].

### 2.5. Construction of the Primary Structure of the Immunogen and Evaluation of Its Characteristics

The final epitopes selected from all the analysis performed were fused with linker peptides such as AAY and GPGPG [28]. These sequences act as ligands for the CTL/B and HTL/B epitopes, respectively. These linkers play a crucial role in protein folding and stabilization and assist in antigenic presentation. Importantly, these ligands have demonstrated efficacy in previous multi-epitope vaccine studies, further reinforcing their utility in the present work [29]. The Heparin-binding hemagglutinin (HBHA) protein expressed by *Mycobacterium* spp. is a necessary TLR4 agonist [30,31] and is known to induce Th1-type responses, aiding in the efficacy of cellular immune responses; for this reason, it was linked to the multi-epitope vaccine through an EAAAK linker to act as an adjuvant. In addition, antigenicity, toxicity, allergenicity, solubility, and physical-chemical parameters were evaluated, respectively, by Vaxijen, ToxinPred, Allertop, and ProtParam programs, and the final sequence was compared via Blastp against the human proteome in the NCBI database to avoid any cross-reaction [32,33,34,35].

### 2.6. Modeling the Secondary and Tertiary Structure of the Multi-Epitope Vaccine

The PSIPRED tool was used to determine the secondary structure of the multi-epitope vaccine [36]. This tool uses an artificial neural network (ANN) and a position-specific scoring matrix (PSSM) to predict secondary structures such as beta-sheets and alpha-helices. The ColabFold server was used to model the tertiary structure [37]. It is a simplified and accelerated version of Alphafold 2, split into two steps: homology search in databases using MMSeqs2 and 3D structure inference using a Python library to communicate with MMSeqs2. The Ramachandran plot was generated through Procheck to assess the quality of the best-generated model by analyzing the amino acids in favorable, additionally allowed, non-favorable, and not allowed regions [38]. Next, the two best models generated by ColabFold were refined by GalaxyRefiner [39], which, to improve the overall quality of the model, reconstructs the side chains, performs the repositioning of the side chains, and subsequently relaxes the structure as a whole. Finally, Procheck was used to re-evaluate the five models created by GalaxyRefine for each protein to choose the one with the best structural quality.

### 2.7. Molecular Docking of the Multi-Epitope Vaccine with Innate Immune System Receptor TLR4

To assess the ability of the vaccine to interact with the innate immune system, we used the HDOCK web [40] tool to perform molecular docking between the vaccine and the TLR4 receptor. Toll-like receptor 4 (PDBID: 3FXI) was downloaded from RCSB (https://www.rcsb.org/ (accessed on 15 May 2023)), followed by removal of all non-TLR structures. The protein–protein docking model with the highest score obtained from HDOCK was selected, and the interactions between receptor and vaccine were evaluated by the PDBepisa tool (https://www.ebi.ac.uk/pdbe/pisa/ (accessed on 15 May 2023)) to predict hydrogen bonds and salt bridges. To cross-check hydrogen bonds and hydrophobic interactions, the LigPlot2 tool was used [41]. In addition, using the HawkDock tool [42], the binding free energy of the docked complex was evaluated (Weng et al., 2019) (http://cadd.zju.edu.cn/hawkdock/ (accessed on 15 May 2023)) using the Molecular Mechanics/Generalized Born Surface Area (MM/GBSA) method. CHIMERA software was used to view the final docking structure [43].

### 2.8. Prediction of Discontinuous B-Cell Epitopes

B-cell epitopes are divided into linear and discontinuous groups based on their conformations. The ElliPro tool [44] was used to predict the discontinuous epitopes suggested for the immunogen (http://tools.iedb.org/ellipro/ (accessed on 15 May 2023)). From the vaccine PDB file, the server uses BLASTp to search for the protein or its homologs in the PDB database. Three algorithms are then run to predict the epitopes. The protein is modeled approximately as an ellipsoid, then the protrusion index of each residue is determined and finally, the surrounding residues are arranged according to their Pl values. Based on the protrusion index, a score is assigned to each probable epitope. The minimum and maximum distances, with standard scores of 0.5 and 6, are the parameters under analysis.

### 2.9. Simulation of the Immune Response of the Multi-Epitope Vaccine

The immune response was simulated by the C-ImmSim server [45] to report the immune response to multi-epitope vaccination more accurately. Three vaccine doses were administered during the simulation, with a four-week interval between each. Each injection contained 1000 multi-epitope proteins. To simulate the three injections, one each week, the time steps were set to 1, 84, and 168 (with each time step equivalent to 8 h in real life, and time step 1 being the time of injection = 0). The incremental steps were modified to 1050, while the other parameters were kept at default values. Two simulations were performed to examine and compare whether the reaction was triggered by the vaccination itself or by the adjuvant sequence alone: (i) the vaccine with adjuvant and (ii) the vaccine alone. In addition, epitope-specific cytokine induction was analyzed for TNF-a, IFN-y, IL17, and IL6 by the Tnepitope [46], IFNepitope [47], IL17eScan [48], and IL6pred [49] tools, which share the use of deep-learning-based algorithms for the prediction of inducing or non-inducing epitopes. Such analyses were performed since TNF-a, IFN-y, IL17, and IL6 are extremely important for the induction of a long-term response and polarization of the cellular response towards Th1/Th17.

### 2.10. Codon Adaptation and In Silico Cloning

To prepare the vaccine for future cloning and expression in an appropriate vector, the in silico cloning process was used. First, the JCat tool [50] was employed to perform codon adaptation in the chimeric vaccine. Using the Escherichia coli K12 expression system, the vaccine sequence was converted into a cDNA sequence. The restriction sites of the *E. coli* pET28a(+) vector, BlpI and BamHI, were introduced to clone the optimized gene sequence. The sequence was inserted into the vector using the SnapGene tool (https://www.snapgene.com/ (accessed on 15 May 2023)) to ensure and visualize potential vaccine expression.

## 3. Results

### 3.1. Prediction, Filtering, and Analysis of CTL, HTL, and B-Cell Epitopes

The sequences of the five antigenic proteins, four from reverse vaccinology and one from the literature (EEH10718.1, EEH11056.1, EEH04925.1, EEH09124.1, and antigen M—AAB84182.2), were retrieved from NCBI databases, and their epitopes were predicted and filtered by IEDB tools for MHCI and MHCII. At the same time, ABCpred performed the prediction of B-cell epitopes. After prediction, filtering by IC50 values and percentile rank reduced the total value of predicted epitopes to 2579 MHCI and 626 MHCII epitopes. After overlapping, clustering, and an immunogenicity analysis (Figure 1), protein EEH10718.1 obtained, respectively, six and five epitopes for MHCI and MHCII, protein EE11056.1 had six and two, EEH04925.1 presented eight and two, EEH09124.1 had two for both, and antigen M harbored only one epitope for MHCI (Appendix A).

Twenty-three MHCI and eleven MHCII epitopes were identified, and all were predicted as B-cell epitopes. AAY linkers linked these epitopes between MHCI epitopes (size 9) and GPGPG peptides linked MHCII epitopes (size 15). In addition, the four previously studied epitopes (“FENLGARL”, “ILGDIGILTNATVFT”, “KPYVLPVPF”, and “FAERIHKLVSLGLNI”) were inserted in this step. Those with more than nine amino acids were inserted in sequence with those of MHCII and the shorter ones with those of MHCI. At the beginning of the protein, the HBHA adjuvant was further linked to the epitopes via the EAAK linker to promote the immune response (Figure 2).

### 3.2. The Final Epitopes Are Shown to Be Highly Conserved among Histoplasma capsulatum Genomes and Present a High Population Coverage

The epitopes of each of the reverse vaccinology proteins and the M antigen were evaluated via the IEDB Epitope Conservancy Analysis tool. The three complete genomes of *H. capsulatum* available in NCBI were used to verify the total, partial, or absence of epitopes (Appendix A). Twenty-one epitopes were entirely conserved in the H88 genome, twenty in the WU24 genome, and thirty-four present entirely in the G186 genome. A total of 14 epitopes are 100% conserved in the three genomes, and there are 11 epitopes with a similarity more significant than 80%, indicating a high conservation overall. The vaccine sequence still has epitopes with a high population coverage, especially in countries with high histoplasmosis prevalence. Countries such as Argentina, Brazil, Canada, Central America, United States, Chile, Colombia, Costa Rica, Ecuador, Mexico, Peru, and Venezuela and European and African countries have 90% or more population coverage by the epitopes (Appendix A). Unfortunately, some specific countries like Paraguay and Guatemala have a range close to 50%.

### 3.3. Prediction of Antigenicity, Allergenicity, Toxicity, and Physicochemical Properties

The final multi-epitope protein was evaluated for antigenicity by the Vaxijen tool, obtaining a score of 0.5957, which is higher than 0.4 (the standard cut-off value for antigenicity). In addition, it has no similarity to any allergenic or toxic protein for humans (Allertop/ToxinPred). Furthermore, it has no similarity to the human proteome. Finally, the protein was proven to be highly stable physicochemically, having a molecular weight of approximately 76,432.07 (76.43 Kda) and an isoelectric point of 5.20. Furthermore, the estimated half-life in humans, yeast, and *E. coli* reticulocytes is, respectively 30, >20, and >10 h. The aliphatic index associated with protein stability in the face of temperature changes was 92.60, indicating a good stability. The significant average of the hydropathy value (GRAVY) is 0.223; these positive values indicate the hydrophobicity of the candidate vaccine. There was no considerable similarity when comparing the final vaccine sequence against the genomes of *Canis lupus familiaris* and *Equus caballus* in the NCBI.

### 3.4. Secondary and Tertiary Structural Properties of the Multi-Epitope Vaccine

The entire amino acid sequence (709 aa) was submitted to PSIPRED. The secondary structure was shown to be mainly composed of alpha helices, with a few coils and beta-strand sequences. We obtained in the fourth model (out of five) the best 3D model generated by ColabFold, which shows 83.4% of amino acid residues in the most favored regions, 11.6% in additional allowed regions, 4.3% of residues in generously allowed regions, and 0.7% in the non-allowed areas in the Ramachandran plot (Appendix A). The refined structure of this protein modeled through the GalaxyWeb server increased the Ramachandran plot scores to 98.3% residues in the most favored regions, 1.3% in additional allowed regions, 0.0% residues in generously allowed regions, and 0.3% in the non-allowed areas (Figure 3).

### 3.5. Docking between the Vaccine and the TLR4 Receptor Shows High Affinity

The HDOCK tool performed protein–protein docking, and the best-predicted model has a docking score of −321.13 and a confidence of 0.9684. Docking analysis revealed that the vaccine binds to TLR4 with twelve hydrogen bonds and eight salt bridges via the PDBepisa tool (Appendix A). Another four hydrogen bonds and using salt bridge were predicted using the ligplot tool. The HawkDock tool further predicted a binding free energy of the complex of −18.13 (kcal/mol). Finally, the docking was visualized by using the Chimera tool (Figure 4).

### 3.6. The “HistoVAC-TSFM” Vaccine Demonstrates High Immunogenicity, Possessing B-Cell Conformational Epitopes, and can Induce the Production of IFN-γ, TNF-α, and Essential Interleukins

The results generated by C-ImmSim are compatible with the accurate and robust development of immunity. The results show that HistoVAC-TSFM can activate and stimulate the natural killer (NK) cells, macrophages, and dendritic cells. We evidenced increased cell types and cytokine release during secondary and tertiary injections. These aspects are essential to promote an efficient immune response, reaching its highest point on the days of application. As expected, a slight reduction in the levels of responsive cells of the innate immune system was found a few days after the third application (Appendix A), accompanied by the potential induction of anti-inflammatory cytokines such as IL-10 and TGF-β. These findings suggest an immune activation without unwanted occurrence of inflammation by the vaccine.

HistoVAC-TSFM seems to be qualified to promote the increase in T helper cells with a remarkable specialization in the Th1 subset. Th1 cells are critical to drive the proliferation of cytotoxic T lymphocytes, thereby amplifying their cytotoxic response capacity. In addition to this effect, it was observed that HistoVAC-TSFM potentially stimulates IFN-γ production, an essential cytokine for the immune response against intracellular pathogens such as *H. capsulatum*. Another foremost finding was the increase in active cytotoxic cells and the reduction in resting cytotoxic cells, which play a crucial role in controlling fungal infections.

In in silico simulations of HistoVAC-TSFM administration, an increase in the memory B cell population over the injection period was observed, with a remarkable specialization in memory B cells and a decrease in the population of non-memory B cells. In addition, during injections, a significant increase in IgM + IgG, IgG1, and IgG1 + IgG2 antibody production was noted. These results highlight the positive effect of vaccination therapy, contributing to a progressive strengthening of the immune response (Figure 5).

We also compared the immune response between the conjugate vaccine with the adjuvant and only the effect of the vaccine without the adjuvant, highlighting that the immune response is sustained by the impact of the vaccine, with specific positive points added by the adjuvant. First, the amount of T and B lymphocytes generated increases minimally when the adjuvant is present. For example, TH (helper) cells during the peak of the second dose exceed 7000 cells when applied with adjuvant, and without adjuvant, they do not reach this value. Also, the total TR (regulatory) cells increases from 180 to 200 when linked with the adjuvant. On the other hand, without the adjuvant, the vaccine reaches a maximum peak of approximately 50 B cells (PLB) in the second dose and declines to less than 50 in the third. When linked with the adjuvant, it not only comes close to reaching the 60-cell mark in the second dose but increases this value in the third dose, indicating an excellent stability (Figure 5 and Figure 6).

Concomitantly, a total of 20 conformational B-cell epitopes were predicted by the Ellipro tool (Appendix A). With a score of 0.922, the best epitope has 17 amino acids, and the one with the lowest score (0.509) has 6 amino acids, demonstrating a very high capacity to generate immune response from antibodies. In addition, 24 epitopes induce TNF-α and IL6, and the Tnepitope, IFNepitope, IL17eScan, and IL6pred tools predicted 13 IFN-γ and IL17. In addition, three epitopes have been shown to induce all four cytokines simultaneously: ‘MLVGINVAV’, ‘FIFAVLLRV’, and ‘ILWDWHPFT’ (Appendix A).

### 3.7. HistoVAC-TSFM Has 10 and 25 MHCI-Inducing Epitopes for Canine and Equine Alleles

Histoplasmosis is not only a serious problem for human public health, but also affects animals that live in frequent contact with them. Dogs and horses are the main animal hosts. From the IEDB “recommended” method for MHCI, we analyzed if any of the epitopes predicted for human alleles would also be present among those predicted for dogs and horses. As a result, we obtained 10 MHCI-inducing epitopes for dogs (Figure 7) and 25 for horses (Appendix A) with a percentile rank lower than 1.

### 3.8. Codon Adaptation and In Silico Cloning of HistoVAC-TSFM

Jcat software analysis for codon adaptation resulted in a 50.73% GC content. The calculated CAI index was 1.0, also within the allowed range. Using SnapGene software, the HistoVAC-TSFM sequence was inserted into the expression vector pET28a (+) (Figure 8).

In summary, starting from four vaccine targets discovered by reverse vaccinology in previous work and from the already well-studied M antigen, 23 epitopes simultaneously induced MHCI and B cells and 11 epitopes of MHCII and B cells were predicted based on IC50 and percentile rank values and python tools for epitope overlap analysis. These epitopes were fused with four already experimentally validated epitopes able to induce strong immune responses against *H. capsulatum*, and the ligands AAY and GPGPG were responsible for the binding between the epitopes. The final vaccine sequence, HistoVAC-TSFM, was analyzed and evaluated for its toxicity, allergenicity, physicochemical parameters, antigenicity, and host non-homology, obtaining highly satisfactory results in all analyses. After that, 2D and 3D vaccine models were developed and refined, and the ability to interact with the TLR4 receptor of the innate immune system was analyzed, obtaining a high confidence value and high docking scores. Finally, in silico immune response simulations, as well as cytokine induction and conformational B-cell epitope analyses, demonstrated that the vaccine has a significant ability to elicit immune responses from CD4+ and CD8+ T lymphocytes, as well as polarize the helper lymphocyte response to a Th1/Th17 pattern through specific cytokines. Finally, the vaccine was cloned in silico in vectors suitable for the gene sequence responsible for its synthesis.

## 4. Discussion

*Histoplasma capsulatum* is a thermally dimorphic fungi that causes histoplasmosis and usually is found in soil enriched with nitrogen, commonly prevenient from bat and bird droppings. Disseminated histoplasmosis is the most severe clinical manifestation and occurs in both immunocompetent and immunocompromised patients, especially HIV-positive individuals, being potentially fatal [51]. Over the last decades, several micro epidemics have occurred, associated with the growth in the world population. Urban growth and deforestation in several countries have increased the exposure of humans to the fungus [13]. Furthermore, behavioral changes in birds and bats are linked to climate change, directly impacting the spread of histoplasmosis. This is supported by the migration from rural areas to urban areas during outbreaks, affecting significantly more people. In the face of the concern about histoplasmosis and in the absence of any existing vaccine, we have designed a multi-epitope vaccine named HistoVAC-TSFM in this work. It is known that the primary adaptive response required to contain and eliminate infection is through T helper lymphocytes (HTLs) of Th1- and Th17-type and cytotoxic T lymphocytes (CTLs) [7,15,52]. Based on this, five vaccine targets already described in the literature had their CTL and HTL epitopes predicted, filtered, and overlapped with B-cell epitopes. Associated with them, four epitopes experimentally validated by previous works were added and were demonstrated to induce a strong immune response in murine animals. The vaccine sequence is highly antigenic and has the necessary characteristics to induce a Th1/Th17 immune response.

T lymphocytes play a critical role in the control and elimination of histoplasmosis. Cytotoxic T lymphocytes (CTLs) produce and secrete perforins, which are responsible for the death of *H. capsulatum*-infected cells. As well as CTLs, T helper lymphocytes (HTLs) control and assist in eliminating the fungus [30]. Th1 polarization of HTLs is essential, since the production of cytokines such as IFN-y and TNF-a responds to primary and secondary infections, respectively [7,15,52]. Thus, a vaccine that provides CTL- and HTL-based immunity with a Th1 pattern is expected to prevent severe clinical manifestations in infected individuals. Another less effective response is the production of antibodies. Previously, antibodies were thought to make no difference in the course of histoplasmosis, but recently, studies using monoclonal antibodies have demonstrated a role in reducing inflammation in the lungs and decreasing the fungal burden [53]. Here, using bioinformatics tools, it was possible to identify epitopes from vaccine targets with a great potential to induce the main immune responses against histoplasmosis. Due to the limitations of a computational approach, these epitopes were filtered and those with the greatest capacity to induce CTLs, HTLs, and B lymphocytes were chosen. However, it is crucial to understand the importance of the vaccine targets included in this approach.

The present study also used vaccine targets and experimentally validated epitopes already described in the literature. The M antigen, an immunodominant glycoprotein of the fungal cell wall, is one of the leading known vaccine targets against *H. capsulatum* [54]. There is also the HSP60 protein that is responsible for the interaction with macrophages and initiates infection. It has been previously used in vaccination strategies, generating robust immune responses with the induction of CD4+ T lymphocytes. However, the high homology with human HSP60 has limited the use of the whole protein as an immunogen [55]. Previous studies demonstrated that four epitopes derived from HSP60, enolase, and HSC82 proteins associated with MHCI and MHCII efficiently induced CD4+ and CD8+ T lymphocyte proliferation and stimulated IFN-γ and IL-17 production [18]. Finally, through reverse vaccinology, four vaccine targets were previously identified by our group. These four targets belong to the core genome of *H. capsulatum*, have no homology with humans, and are extracytoplasmic [19]. Thus, these four vaccine targets, in addition to the M antigen and the four experimentally validated epitopes, were used to construct a new multi-epitope vaccine called HistoVAC-TSFM.

By immunoinformatic, 34 epitopes were predicted from the five proteins analyzed: 23 from MHCI and 11 from MHCII. The vaccine was developed using the 38 epitopes (4 already experimentally validated) and was shown to be highly antigenic and without toxicity or allergenicity to humans. A comparison of the human proteome with the multi-epitope protein showed no similarity, which is an essential analysis since vaccines for eukaryotic microorganisms can have similarities with humans, causing adverse reactions. This indicates that the two HSP60 epitopes tested in murine animals do not have similarities with humans, overcoming the difficulty in generating HSP60-based vaccines without adverse effects from host similarity. Furthermore, protein–protein docking between the multi-epitope vaccine and the TLR4 immune receptor demonstrated a high capacity of interaction, with 10 hydrogen bonds occurring and potentially inducing the receptor, leading concomitantly to the induction of pro-inflammatory cytokines and increasing the capacity of phagocytosis of host cells. Increased pro-inflammatory cytokines such as IFN-y, for example, can induce dendritic cell maturation, activate them, and facilitate TLR signaling pathways when administered concomitantly with TLR agonists [56]. Here, the results obtained by docking suggest the vaccine’s ability to induce an initial pro-inflammatory response via the TLR4 receptor, followed by the maturation of adaptive immune response cells.

Since Th1 and Th17 responses are primarily for containment and elimination of the pathogen, being able to polarize CD4+ T lymphocytes to this pattern is the main objective of a vaccine against *H. capsulatum*. Th1 cells produce large amounts of IFN-y, a cytokine with several functions associated with the containment of the load and elimination of *H. capsulatum*. In addition to activating phagocytes, they stimulate murine macrophages to produce nitric oxide and regulate the amount of iron and zinc available to the fungus, limiting intracellular proliferation. In addition to these effector mechanisms, TNF-a, also secreted by lymphocytes, has a critical function, regulating the emergence of regulatory T cells (Tregs) and the magnitude of apoptosis. Patients taking TNF-a inhibitors, for example, present a serious risk of developing disseminated histoplasmosis, demonstrating how pertinent this cytokine is for immunity [57]. Concomitantly, in the presence of IL6, CD4+ cells differentiate into Th17 cells, the leading producers of IL17. This interleukin induces the production of TNF-a and IL-1β, as well as CXC chemokines involved in the recruitment and activation of Mϕ and neutrophils in various models of infection [58,59].

The vaccine designed in this study harbors 23 MHCI epitopes responsible for developing long-term immunity through cytotoxic T lymphocytes. These epitopes induce these cells to eliminate the pathogen from tissue through their mechanisms, such as perforins. It is also expected that helper T lymphocytes will develop through the 11 predicted MHCII epitopes to ensure immunity to vaccinated individuals. Our data suggest that the HistoVAC-TSFM vaccine can lead to the synthesis of IFN-y, TNF-a, IL6, and IL17. For example, three epitopes were predicted to induce all four cytokines simultaneously. In this way, CD4+ T lymphocytes would be able to differentiate into the Th1 and Th17 pattern, generating the cytokines essential for pathogen containment and elimination. In addition, the secondary and tertiary structure of the vaccine has high structural quality, with the vast majority of amino acids in favorable regions, which favors the synthesis of the protein in silico. The tertiary structure was further evaluated for B-cell conformational epitopes, where 19 epitopes were identified. Although B cells and antibodies do not have a central role in the response against histoplasmosis, they are known to be complementary and assist in reducing the fungal burden [60].

The vaccine was also conjugated with the adjuvant HBHA, an essential inducer of TLR4, and through the C-immsim tool, we simulated the immune response. The tool showed that the vaccine does not need the adjuvant to promote the primary immune responses, such as antibody synthesis or an increase in the population of T and B cells. Nevertheless, the responses maintained a more significant population of active cells throughout the three doses. In addition, central cytokines such as IFN-y and TNF-a are generated in large amounts by the three doses, and the primary responses necessary for the containment and elimination of the pathogen are established (HTLs and CTLs) [30].

Histoplasmosis still occurs with a high frequency in animals such as dogs and horses. In dogs, the infection may be subclinical and cause clinical pulmonary granulomatous disease or disseminated disease. The disseminated disease affects several organs, such as the liver, spleen, gastrointestinal tract, bone and bone marrow, integument, and eyes [61]. Equine histoplasmosis, commonly known as epizootic lymphangitis (EL), is a neglected granulomatous disease in these animals and affects African countries like Ethiopia, but there are reports from India, Iran, Pakistan, and Japan [62,63]. Animals often show thickening of lymphatic vessels and nodes, leading to suppurative and ulcerative dermatitis and lymphangitis, affecting animal welfare and the livelihood of cart-owning families. Since no vaccines exist for humans or animals, we assessed whether epitopes predicted for human alleles could possess any immunogenicity for canine and equine alleles. Finally, we identified 10 epitopes for dogs and 25 for horses, which could be used in new studies for the development of vaccines for animals or to verify whether the HistoVAC-TSFM vaccine containing such epitopes would be sufficient to induce a protective response in both humans and these animals [64].

Future in vitro and in vivo analyses are needed to confirm the immunological potential of the proposed vaccine; however, given the existing difficulties in combating histoplasmosis, this is the beginning in the development of a new immunogen capable of preventing the disease in humans and animals. Multi-epitope vaccines generally overcome the problems of vaccines based on a single antigen or whole-cell vaccines. These vaccines manage to elicit the immune system against different proteins of the pathogen, thus preventing genetic events such as mutations from causing a loss of efficacy in vaccines based on a single antigen, while maintaining a low capacity to provoke allergenic responses or toxic reactions that whole-cell vaccines can generate [65]. As a neglected fungal disease, histoplasmosis is often confounding with other infections, especially tuberculosis, community-acquired pneumonia, sarcoidosis, and others [66,67]. Despite culture remaining the gold standard of diagnosis, it can take up to 6–8 weeks to grow and the sensitivity is poor [68]. Therefore, without an accurate diagnosis, the treatment can take a long time to begin, increasing the risk of death of the infected patient.

Two of the main challenges associated with this vaccine are population coverage and non-homology to the host. Using the latest computational approaches, it is possible to predict a broad population coverage for various countries on different continents, such as the Americas, Europe, and Africa. The vaccine also uses four experimentally validated epitopes that overcome the similarities and possible side effects caused by eukaryotic similarity. It is hoped that by using subtractive genomics, the other previously predicted vaccine targets will not cause unwanted responses, giving the immunogen specificity. In our approach, it is expected that the vaccine will be able to act by preventing severe cases of histoplasmosis, inducing a robust immune response without possible allergenic or toxic secondary effects; however, to confirm this, new trials with animal models immunized with HistoVAC-TSFM are necessary.

## 5. Conclusions

Historically, *H. capsulatum* was the primary pathogen associated with bat feces, cave explorers, and chicken farmers. The entire population, regardless of age and location, is predisposed to be infected if they inhale a considerable number of fungal cells. Immunosuppressed individuals especially present severe symptoms and a high mortality. There are currently no vaccines for histoplasmosis. We propose in this study a vaccine based on 38 epitopes from vaccine targets predicted by reverse vaccinology or in previous experimental studies. The vaccine can induce long-term immune responses based on cytotoxic and helper B- and T-cell epitopes, potentially inducing Th1 and Th17 patterns and cytokines such as IFN-y, TNF-a, IL17, and IL6. HistoVAC-TSFM was shown to be highly antigenic, non-toxic, and non-allergenic and bears no resemblance to human proteins. Although four epitopes have previously been experimentally validated in murine models, further in vitro and in vivo studies are required to confirm the immunogenicity of the recombinant protein in humans.

## Figures and Tables

**Figure 1 jof-10-00043-f001:**
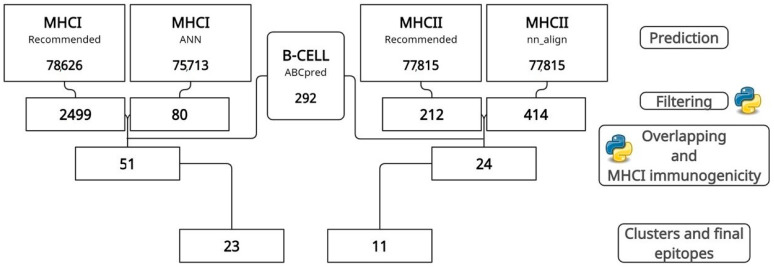
Step-by-step prediction, filtering, and overlapping of epitopes by IEDB and ABCpred tools. In the end, 23 MHCI and 11 MHCII epitopes were selected.

**Figure 2 jof-10-00043-f002:**
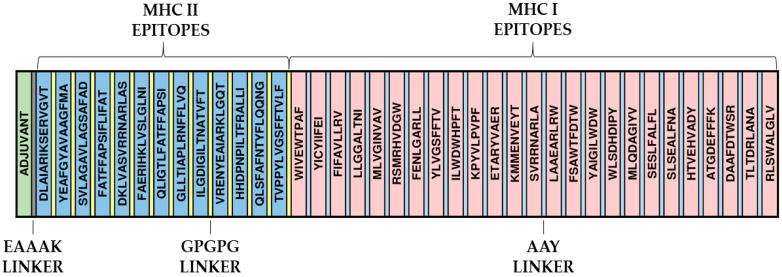
Assembly of the final sequence of the multi-epitope vaccine and adjuvant. The EAAAK linker appears between adjuvant and multi-epitope sequences, while MHCI and MHCII epitopes have AAY GPGPG peptides as linkers.

**Figure 3 jof-10-00043-f003:**
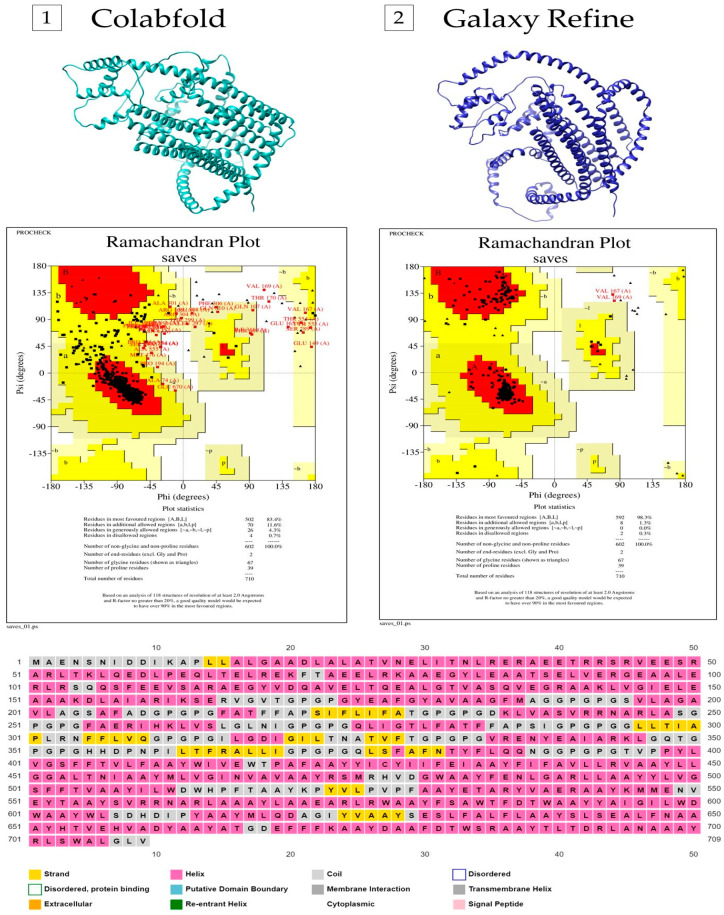
Secondary and tertiary structures of the multi-epitope vaccine. On the left side (number 1), there is the structure of the protein predicted by ColabFold, as well as the corresponding Ramachandran plot. On the right side (number 2), there is the structure of the protein refined by GalaxyRefiner and its Ramachandran plot. Below is the secondary structure predicted by PsiPred.

**Figure 4 jof-10-00043-f004:**
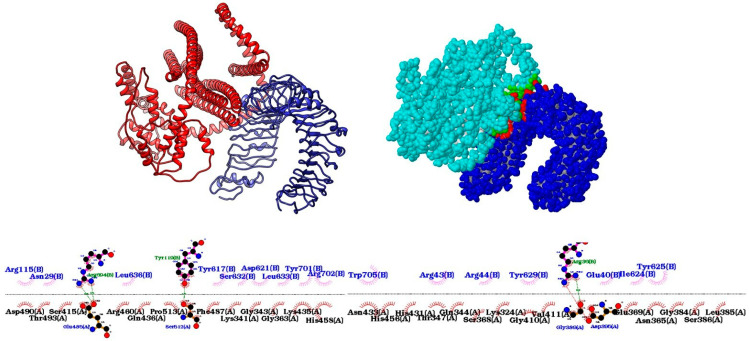
Visualization of protein–protein docking. At the top, there is the docking visualized by the Chimera tool (vaccine in red and TLR4 in blue), and the contact regions by PDBepisa (vaccine in light blue and TLR4 in dark blue), respectively. Below are the interactions using the Ligplot tool.

**Figure 5 jof-10-00043-f005:**
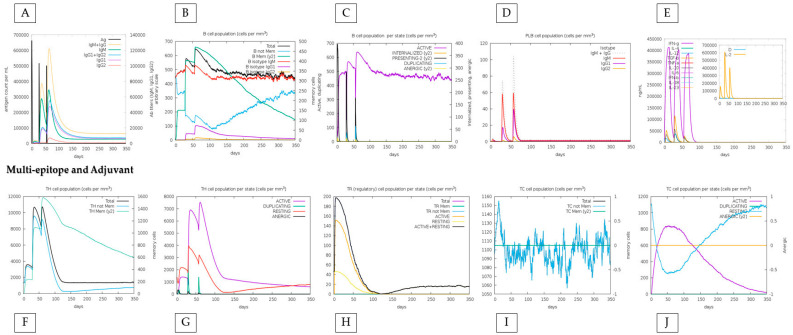
In silico immune response simulation results by C-ImmSim for the adjuvant-linked vaccine. (**A**) a simulation of the production of immunoglobulins such as IgG and IgM, (**B**) the population of B cells, (**C**) the population of B cells by state (active or not), (**D**) the production of PLB cells, (**E**) the production of cytokines. In the graphs at the bottom of the figure, (**F**) shows the population of TH cells, (**G**) shows TH cells by state (active or not), (**H**) shows the population of Treg cells, and (**I**,**J**) show the population and state of cytotoxic T cells.

**Figure 6 jof-10-00043-f006:**
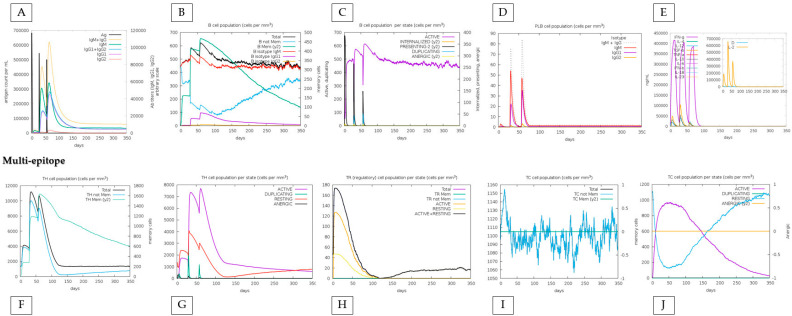
In silico immune response simulation results by C-ImmSim only for the multi-epitope vaccine. The analysis was carried out in the same order as in Figure 5. (**A**–**D**) represent the responses associated with B cells; (**E**) shows the graph of cytokine production; (**F**,**G**) represent TH cells; (**H**) shows the diagram of Treg cell populations; and (**I**,**J**) represent populations and states of cytotoxic T cells.

**Figure 7 jof-10-00043-f007:**
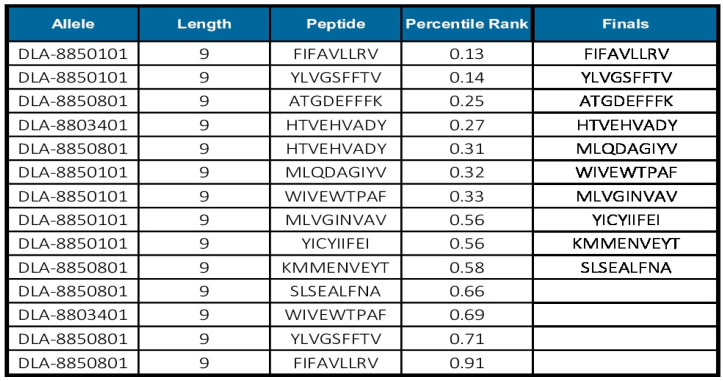
Prediction of MHCI epitopes using IEDB tools for canine alleles. Ten MHCI epitopes for humans are coincidentally excellent epitopes for canine alleles.

**Figure 8 jof-10-00043-f008:**
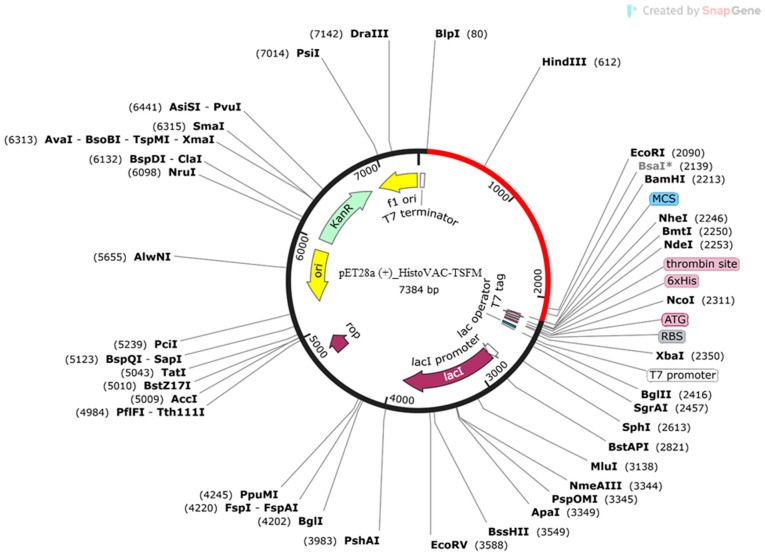
The insertion of the HistoVAC-TSFM sequence is present in red using the BamHI and BlpI restriction enzymes.

## Data Availability

Data are contained within the article and Appendix A.

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
