# Peer review of "Design of a Multi-Epitope Vaccine against Histoplasma capsulatum through Immunoinformatics Approaches"

_jof, 2024, doi:10.3390/jof10010043_

Round 1
Reviewer 1 Report
Comments and Suggestions for Authors
In the article “Design of a multi-epitope vaccine against Histoplasma capsulatum through immunoinformatics approaches,” the authors use immunoinformatics as a novel approach to determine the main immunogenic components of antigens through in silico methods for the design of a multi-epitope vaccine called HistoVAC-TSFM, capable of promoting an effective and long-lasting T lymphocyte immune response with probable induction of Th1 and Th17 patterns essential for preventing histoplasmosis.
It is a well-written, well-planned article; however, I have a few comments:
Histoplasma strains that have been used to identify antigens and epitopes capable of promoting an effective and long-lasting immune response against histoplasmosis generally use the reference strain G-217B that has been genotyped and classified in the phylogenetic group NAm2 and has been proposed to be renamed as a cryptic species Histoplasma ohiense, as well as other strains from the United States. Will all phylogenetic species share these epitopes? It would be interesting if the authors also discussed that several phylogenetic and new cryptic species distributed worldwide are currently considered.
Line 163. Regarding the genotypic variability of this fungus, the authors considered that new variants may arise due to mutational and recombination events. Therefore, the authors verified the epitope coverage among the H. capsulatum strains using the IEDB tool "Epitope Conservancy Analysis" together with the complete reference genomes of the species (H88, WU24, and G186). The authors should explain which phylogenetic species represent the complete genomes of species H88, WU24, and G186.
References
Uniform references according to the style of the journal.
Author Response
##Reviewer 1
In the article “Design of a multi-epitope vaccine against Histoplasma capsulatum through immunoinformatics approaches,” the authors use immunoinformatics as a novel approach to determine the main immunogenic components of antigens through in silico methods for the design of a multi-epitope vaccine called HistoVAC-TSFM, capable of promoting an effective and long-lasting T lymphocyte immune response with probable induction of Th1 and Th17 patterns essential for preventing histoplasmosis. It is a well-written, well-planned article; however, I have a few comments:
1. Histoplasma strains that have been used to identify antigens and epitopes capable of promoting an effective and long-lasting immune response against histoplasmosis generally use the reference strain G-217B that has been genotyped and classified in the phylogenetic group NAm2 and has been proposed to be renamed as a cryptic species Histoplasma ohiense, as well as other strains from the United States. Will all phylogenetic species share these epitopes? It would be interesting if the authors also discussed that several phylogenetic and new cryptic species distributed worldwide are currently considered.
Answer to reviewer: First of all, we would like to thank the reviewer 1 for the positive and constructive comments about the manuscript. The vaccine proposed in this work is focused only on protection against the species H. capsulatum, due to its relevance and impact in worldwide. We agree that other pathogenic fungi phylogenetically related can contain or not the epitopes predicted here, however, as we do not specifically intend in this work to generate new forms of prevention against other species, such as H. ohiense, we did not verify the conservation of epitopes for them. Additionally, the species H. ohiensedid not have complete genomes available at NCBI. Thus, using contigs to verify the presence or absence of epitopes could generate false negatives, given the uncertainty of whether a sequence is actually present or not in the complete genome.
2. Line 163. Regarding the genotypic variability of this fungus, the authors considered that new variants may arise due to mutational and recombination events. Therefore, the authors verified the epitope coverage among the H. capsulatumstrains using the IEDB tool "Epitope Conservancy Analysis" together with the complete reference genomes of the species (H88, WU24, and G186). The authors should explain which phylogenetic species represent the complete genomes of species H88, WU24, and G186.
Answer to reviewer: Thank you for your comments. The genomes of Histoplasma capsulatum used in this work “H88, WU24 and G186” are the only three complete genomes available at NCBI. Our aim was to verify the presence or absence of the predicted epitopes in these three complete genomes, of this particular species (H. capsulatum). This information was included in the text (line 388). Furthermore, the genomes used were identified in three different nationalities (USA, Panama and Belgium). Therefore, the analysis of conservation was used to predict how much the vaccine could cover the strains of H. capsulatum worldwide, maximizing the chances of the epitopes included in the vaccine being effective against other strains not yet sequenced, from different locations. To reliably check the presence of epitopes in new species such as H. ohiense and H. mississippiense, would be required the complete sequenced genomes available at NCBI, such as exist for the species H. capsulatum. In the future, with new sequencing being carried out, we will be able to carry out new work on the conservation of epitopes in different species of Histoplasma.
Reviewer 2 Report
Comments and Suggestions for Authors
The work presented by Marques and collaborators is very pertinent and necessary, as it provides significant data for the development of a vaccine for this important mycosis with worldwide distribution.
The work is very well designed, and the results are consistent with the materials and methods used. I consider that the work can be published, however it has aspects that must be improved.
Introduction
The introduction is very long and has aspects that are not necessary and that do not support the title of the manuscript.
The information provided related to the clinical manifestations of histoplasmosis is not relevant.
Paragraph 3 of the introduction (lines 70 – 74) is a bit contradictory, as it begins by mentioning that histoplasmosis is underdiagnosed, but ends by stating that it is well documented in Brazil. Do the authors consider this information relevant to the title of the work?
The introduction could be more concise and omit information not relevant to the work. But it could include a small paragraph in which the difficulties for the development of fungal vaccines are related, even considering aspects of immunoinformatics.
Materials and Methods
The materials and methods are well described, clear and coherent.
Results
The results described in the text are very good and consistent with the methods used, and they also answer the question implicit in the title of the work. However, figures 5 and 6, which I consider to be very relevant, are not very clear, it is difficult to understand and interpret so many saturated images. I consider it necessary to build the images again.
Discussion
The second paragraph of the discussion (lines 484 - 495) does not discuss in a clear and concise manner the results obtained in the work, which is why it should be removed or constructed again, trying to include in it the discussion of some results.
The third paragraph of the discussion (lines 496 - 512) the discussion is very limited to what is described only in line 504 of this paragraph, the rest is a theoretical support that can be very important but is not part of the discussion of no work results.
Paragraph 5 of the discussion (lines 543 - 557) is appropriate, as it comes close to a clear discussion where the impact of the results is clearly demonstrated.
In general terms, the discussion is very long and includes many theoretical aspects that are not directly linked to the results.
In addition to the above, the discussion must be improved, it must make a real analysis of the results and their impact on the development of vaccines, it must also include an analysis of the difficulties of this process.
Author Response
##Reviewer 2
The work presented by Marques and collaborators is very pertinent and necessary, as it provides significant data for the development of a vaccine for this important mycosis with worldwide distribution. The work is very well designed, and the results are consistent with the materials and methods used. I consider that the work can be published, however it has aspects that must be improved.
1- Introduction
The introduction is very long and has aspects that are not necessary and that do not support the title of the manuscript. The information provided related to the clinical manifestations of histoplasmosis is not relevant. Paragraph 3 of the introduction (lines 70 – 74) is a bit contradictory, as it begins by mentioning that histoplasmosis is underdiagnosed, but ends by stating that it is well documented in Brazil. Do the authors consider this information relevant to the title of the work?
Answer to reviewer: Thank you for your comments. The introduction was reduced when possible. We believe that the information about the clinical manifestations is relevant since it can introduce the disease in animals and justify the analysis of epitopes in animals done in this paper.
The central idea of ​​paragraph 3 was to explain that cases of histoplasmosis are underdiagnosed considering the difficulty of diagnosis, but the presence of the pathogen in several countries, such as the USA and Brazil, is well documented. The sentence was reformulated in the text. We consider important this information, once the disease is neglected in several aspects and is necessary to develop different ways to prevent this important infection, especially in immunocompromised patients.
3- The introduction could be more concise and omit information not relevant to the work. But it could include a small paragraph in which the difficulties for the development of fungal vaccines are related, even considering aspects of immunoinformatics.
Answer to reviewer: The previous sentences 55-58 and 102-105 of introduction were excluded and the previous lines 108-112 were summarized. Additionally, we include a small paragraph at the end of introduction about the difficulties for the development of fungal vaccines.
4- The results described in the text are very good and consistent with the methods used, and they also answer the question implicit in the title of the work. However, figures 5 and 6, which I consider to be very relevant, are not very clear, it is difficult to understand and interpret so many saturated images. I consider it necessary to build the images again.
Answer to reviewer: The figures 5 and 6 were made by the C-immsim program. Therefore, it is not possible to modify them, unfortunately. However, both figures have the minimum resolution required by the JoF.
5- The second paragraph of the discussion (lines 484 - 495) does not discuss in a clear and concise manner the results obtained in the work, which is why it should be removed or constructed again, trying to include in it the discussion of some results.
Answer to reviewer: Thank you for your comments. We agree with you and the paragraph was excluded.
6- The third paragraph of the discussion (lines 496 - 512) the discussion is very limited to what is described only in line 504 of this paragraph, the rest is a theoretical support that can be very important but is not part of the discussion of no work results.
Answer to reviewer: We agree with you. The changes have been made in this paragraph.
7- Paragraph 5 of the discussion (lines 543 - 557) is appropriate, as it comes close to a clear discussion where the impact of the results is clearly demonstrated. In general terms, the discussion is very long and includes many theoretical aspects that are not directly linked to the results. In addition to the above, the discussion must be improved, it must make a real analysis of the results and their impact on the development of vaccines, it must also include an analysis of the difficulties of this process.
Answer to reviewer: Thank you for your comments. The changes have been made. We improved the discussion, included some information linking it to our results and excluded some sentences that we considered did not change the understanding of the text.
Reviewer 3 Report
Comments and Suggestions for Authors
The paper effectively addresses the need for a vaccine against Histoplasma capsulatum, emphasizing the disease's prevalence, morbidity, and mortality rates and providing a clear context for the research. The multi-epitope vaccine (HistoVAC-TSFM) design is comprehensive, incorporating 38 epitopes with strategic distribution among CTL, HTL, and B cell responses and the inclusion of validated epitopes from the B subunit of cholera toxin as an adjuvant.
Thorough computer tests show that the vaccine is stable, doesn't harm or cause allergies, and doesn't share any properties with the host. This gives people faith in the proposed vaccine's safety profile. The paper effectively communicates the vaccine's potential to induce both innate and adaptive immune responses, including the activation of cytotoxic and helper B and T cell epitopes and the likelihood of Th1 and Th17 patterns.
The vaccine's ability to induce significant cytokines, such as IFN-γ, TNF-α, IL17, and IL6, is highlighted, demonstrating a comprehensive immune response that could be crucial in combating histoplasma infections.
The confirmation that the vaccine is highly antigenic, non-toxic, and non-allergenic, along with its dissimilarity to human proteins, supports its potential as a safe and effective immunization strategy.
Recognizing the need for more in vitro and in vivo studies to confirm that the recombinant protein can activate the immune system in humans shows that the proposed vaccine is being developed in a realistic and careful way.
On the other hand, while the historical context of Histoplasma capsulatum is mentioned, the paper could delve deeper into the historical challenges of addressing histoplasmosis and the limitations of previous approaches, providing a more thorough background.
The paper mentions that symptoms of histoplasmosis are frequently mistaken for other diseases. A more detailed exploration of the challenges associated with clinical diagnosis and the potential consequences of misdiagnosis would enhance the discussion.
The statement about the slow growth of the fungus in microbiological diagnostics could benefit from additional information on how this characteristic poses challenges and potential implications for timely diagnosis and treatment.
The paper lacks a discussion on ethical considerations related to vaccine development, such as potential side effects or unintended consequences, which is crucial for a comprehensive analysis.
The paper mentions the prevalence of histoplasmosis in the Americas but does not discuss the potential applicability of the vaccine to other regions where the disease might also pose a threat.
While the paper discusses the non-homology with the host, it could provide more details on the specificity of the vaccine and its potential impact on off-target effects.
Author Response
##Reviewer 3
The paper effectively addresses the need for a vaccine against Histoplasma capsulatum, emphasizing the disease's prevalence, morbidity, and mortality rates and providing a clear context for the research. The multi-epitope vaccine (HistoVAC-TSFM) design is comprehensive, incorporating 38 epitopes with strategic distribution among CTL, HTL, and B cell responses and the inclusion of validated epitopes from the B subunit of cholera toxin as an adjuvant. Thorough computer tests show that the vaccine is stable, doesn't harm or cause allergies, and doesn't share any properties with the host. This gives people faith in the proposed vaccine's safety profile. The paper effectively communicates the vaccine's potential to induce both innate and adaptive immune responses, including the activation of cytotoxic and helper B and T cell epitopes and the likelihood of Th1 and Th17 patterns.
The vaccine's ability to induce significant cytokines, such as IFN-γ, TNF-α, IL17, and IL6, is highlighted, demonstrating a comprehensive immune response that could be crucial in combating histoplasma infections. The confirmation that the vaccine is highly antigenic, non-toxic, and non-allergenic, along with its dissimilarity to human proteins, supports its potential as a safe and effective immunization strategy. Recognizing the need for more in vitro and in vivo studies to confirm that the recombinant protein can activate the immune system in humans shows that the proposed vaccine is being developed in a realistic and careful way.
On the other hand, while the historical context of Histoplasma capsulatum is mentioned, the paper could delve deeper into the historical challenges of addressing histoplasmosis and the limitations of previous approaches, providing a more thorough background.
The paper mentions that symptoms of histoplasmosis are frequently mistaken for other diseases. A more detailed exploration of the challenges associated with clinical diagnosis and the potential consequences of misdiagnosis would enhance the discussion. The statement about the slow growth of the fungus in microbiological diagnostics could benefit from additional information on how this characteristic poses challenges and potential implications for timely diagnosis and treatment.
Answer to reviewer: Thank you for your comments. We agree with you and more information was added in the text to clarify this point (958-971 lines).
2- The paper lacks a discussion on ethical considerations related to vaccine development, such as potential side effects or unintended consequences, which is crucial for a comprehensive analysis. The paper mentions the prevalence of histoplasmosis in the Americas but does not discuss the potential applicability of the vaccine to other regions where the disease might also pose a threat. While the paper discusses the non-homology with the host, it could provide more details on the specificity of the vaccine and its potential impact on off-target effects.
Answer to reviewer: We would like to thank the reviewer 3 for the comments. The changes were done as suggested. We include some considerations in the last paragraph of the discussion.